# CycleMLP: A MLP-like Architecture for Dense Prediction

**Shoufa Chen[1]**     **Enze Xie[1]**     **Chongjian Ge[1]**     **Runjian Chen[1]**     **Ding Liang[2]**     **Ping Luo[1, 3]**
[1] The University of Hong Kong     [2] SenseTime Research
[3] Shanghai AI Laboratory, Shanghai, China
{shoufach, xieenze, rhettgee, rjchen}@connect.hku.hk
liangding@sensetime.com   pluo@cs.hku.hk

## Abstract

This paper presents a simple MLP-like architecture, CycleMLP, which is a versatile backbone for visual recognition and dense predictions. As compared to modern MLP architectures, *e.g.* , MLP-Mixer (Tolstikhin et al., 2021), ResMLP (Touvron et al., 2021a), and gMLP (Liu et al., 2021a), whose architectures are correlated to image size and thus are infeasible in object detection and segmentation, CycleMLP has two advantages compared to modern approaches. (1) It can cope with various image sizes. (2) It achieves linear computational complexity to image size by using local windows. In contrast, previous MLPs have $O(N^2)$ computations due to fully spatial connections. We build a family of models which surpass existing MLPs and even state-of-the-art Transformer-based models, *e.g.* Swin Transformer (Liu et al., 2021b), while using fewer parameters and FLOPs. We expand the MLP-like models' applicability, making them a versatile backbone for dense prediction tasks. CycleMLP achieves competitive results on object detection, instance segmentation, and semantic segmentation. In particular, CycleMLP-Tiny outperforms Swin-Tiny by 1.3% mIoU on ADE20K dataset with fewer FLOPs. Moreover, CycleMLP also shows excellent zero-shot robustness on ImageNet-C dataset. Code is available at https://github.com/ShoufaChen/CycleMLP.

## 1 Introduction

Vision models in computer vision have been long dominated by convolutional neural networks (CNNs) (Krizhevsky et al., 2012; He et al., 2016). Recently, inspired by the successes in Natural Language Processing (NLP) field, Transformers (Vaswani et al., 2017) are adopted into the computer vision community. Built with self-attention layers, multi-layer perceptrons (MLPs), and skip connections, Transformers make numerous breakthroughs on visual tasks (Dosovitskiy et al., 2020; Liu et al., 2021b). More recently, (Tolstikhin et al., 2021; Liu et al., 2021a) have validated that building models solely on MLPs and skip connections without the self-attention layers can achieve surprisingly promising results on ImageNet (Deng et al., 2009) classification.

Despite promising results on visual recognition tasks, these MLP-like models can not be used in dense prediction tasks (*e.g.,* object detection and semantic segmentation) due to the three challenges: **(1)** Current models are composed of blocks with non-hierarchical architectures, which make the model infeasible to provide pyramid and high-resolution feature representations. **(2)** Current models cannot deal with flexible input scales due to the Spatial FC as shown in Figure 1b. The spatial FC is configured by an image-size related weight[1]. Thus, this

| FC | Stepsize | $\mathcal{O}(HW)$ | Scale Variable | ImgNet Top-1 | COCO AP | ADE20K mIoU |
|---|---|---|---|---|---|---|
| Channel | 1 | $HW$ | ✓ | 79.4 | 35.0 | 36.3 |
| Spatial | - | $H^2W^2$ | ✗ | 80.9 | ✗ | ✗ |
| Cycle | 7 | $HW$ | ✓ | 81.6 | 41.7 | 42.4 |

Table 1: Comparison of three types of FC operators.

---

[1]We omit *bias* here for discussion convenience.

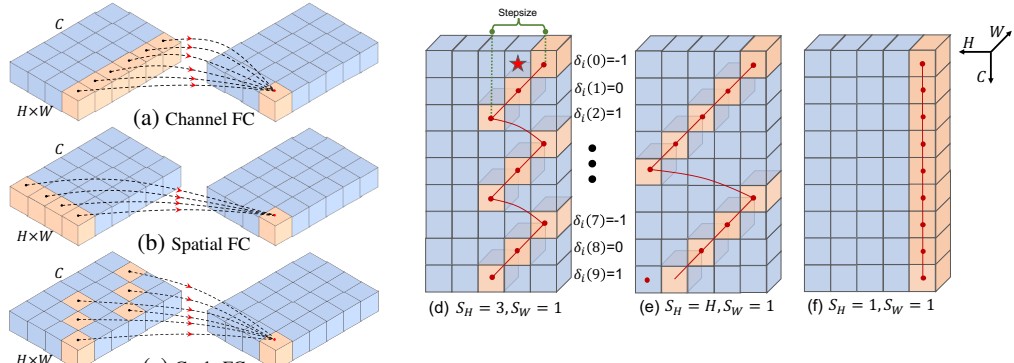

Figure 1: **(a)-(c): motivation of Cycle Fully-Connected Layer (Cycle FC)** compared to Channel FC and Spatial FC. **(a)** Channel FC aggregates features in the channel dimension with spatial size '1'. It can handle various input scales but cannot learn spatial context. **(b)** Spatial FC (Tolstikhin et al., 2021; Touvron et al., 2021a; Liu et al., 2021a) has a global receptive field in the spatial dimension. However, its parameter size is fixed and it has quadratic computational complexity to image scale. **(c)** Our proposed Cycle Fully-Connected Layer (Cycle FC) has linear complexity the same as channel FC and a larger receptive field than Channel FC. **(d)-(f): Three examples of different stepsizes.** Orange blocks denote the sampled positions. ★ denotes the output position. For simplicity, we omit batch dimension and set the feature's width to 1 here for example. Several more general cases can be found in Figure 7 (Appendix G). Best viewed in color.

structure typically requires the input image with a fixed scale during both the training and inference procedure. It contradicts the requirements of dense prediction tasks, which usually adopt a multi-scale training strategy (Carion et al., 2020) and different input resolutions in training and inference stages (Lin et al., 2014; Cordts et al., 2016). **(3)** The computational and memory costs of the current MLP models are quadratic to input image sizes for dense prediction tasks (e.g., COCO benchmark (Lin et al., 2014)).

To address the first challenge, we construct a hierarchical architecture to generate pyramid features. For the second and third issues, we propose a novel variant of fully connected layer, named as *Cycle Fully-Connected Layer (Cycle FC)*, as illustrated in Figure 1c. The Cycle FC is capable of dealing with various image scales and has linear computational complexity to image size.

Our Cycle FC is inspired by Channel FC layer illustrated in Figure 1a, which is designed for channel information communication (Lin et al., 2013; Szegedy et al., 2015; He et al., 2016; Howard et al., 2017). The main merit of Channel FC lies in that it can deal with flexible image sizes since it is configured by image-size agnostic weight of $C_{in}$ and $C_{out}$. However, the Channel FC is infeasible to aggregate spatial context information due to its limited receptive field.

Our Cycle FC is designed to enjoy Channel FC's merit of taking input with arbitrary resolution and linear computational complexity while enlarging its receptive field for context aggregation. Specifically, Cycle FC samples points in a cyclical style along the channel dimension (Figure 1c). In this way, Cycle FC has the same complexity (both the number of parameters and FLOPs) as channel FC while increasing the receptive field simultaneously. To this end, we adopt Cycle FC to replace the Spatial FC for spatial context aggregation (*i.e.,* token mixing) and build a family of MLP-like models for both recognition and dense prediction tasks.

The contributions of this paper are as follows: **(1)** We propose a new MLP-like operator, Cycle FC, which is computational friendly to cope with flexible input resolutions. **(2)** We take the first attempt to build a family of hierarchical MLP-like architectures (CycleMLP) based on Cycle FC operator for dense prediction tasks. **(3)** Extensive experiments on various tasks (e.g., ImageNet classification, COCO object instance detection, and segmentation, and ADE20K semantic segmentation) demonstrate that CycleMLP outperforms existing MLP-like models and is comparable to and sometimes better than CNNs and Transformers on dense predictions.

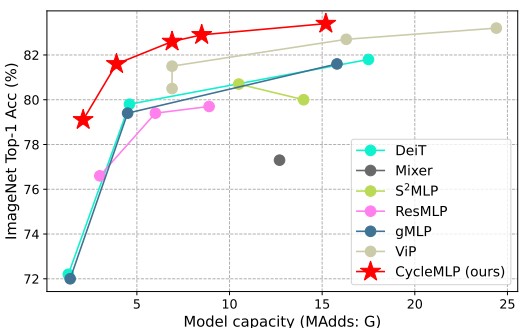

Figure 2: **ImageNet accuracy *v.s. model capacity.*** All models are trained on ImageNet-1K (Deng et al., 2009) without extra data. CycleMLP surpasses existing MLP-like models such as MLP-Mixer (Tolstikhin et al., 2021), ResMLP (Touvron et al., 2021a), gMLP (Liu et al., 2021a), S$^2$-MLP (Yu et al., 2021) and ViP (Hou et al., 2021).

**Related Work.** Convolution Neural Networks (CNNs) has dominated the visual backbones for several years (Krizhevsky et al., 2012; Simonyan & Zisserman, 2014; He et al., 2016). (Dosovitskiy et al., 2020) introduced the first pure Transformer-based (Vaswani et al., 2017) model into computer vision and achieved promising performance, especially pre-trained on the large scale JFT dataset. Recently, some works (Tolstikhin et al., 2021; Touvron et al., 2021a; Liu et al., 2021a) removed the attention in Transformer and proposed pure MLP-based models. Please see Appendix A for a comprehensive review of the literature on the visual backbones.

## 2 METHOD

In this section, we introduce CycleMLP models for vision tasks including recognition and dense predictions. To begin with, in Sec. 2.1 we formulate our proposed novel operator, Cycle FC, which serves as a basic component for building CycleMLP models. Then we compare Cycle FC with Channel FC and multi-head attention adopted in recent Transformer-based models (Dosovitskiy et al., 2020; Touvron et al., 2020; Liu et al., 2021b) in Sec. 2.2. Finally, we present the detailed configurations of CycleMLP models in Sec. 2.3.

### 2.1 CYCLE FULLY-CONNECTED LAYER

**Notation.** We denote an input feature map as $\boldsymbol{X} \in \mathbb{R}^{H \times W \times C_{in}}$, where $H, W$ denote the height and width of the image and $C_{in}$ is the number of feature channels. We use subscripts to index the feature map. For example, $\boldsymbol{X}_{i,j,c}$ is the value of $c^{th}$ channel at the spatial position $(i, j)$ and $\boldsymbol{X}_{i,j,:}$ are values of all channels at the spatial $(i, j)$.

**The motivation** behind Cycle FC is to enlarge receptive field of MLP-like models to cope with downstream dense prediction tasks while maintaining the computational efficiency. As illustrated in Figure 1a, Channel FC applies weighting matrix on $\boldsymbol{X}$ along the channel dimension on fixed position $(i, j)$. However, Cycle FC introduces a receptive field of $(S_H, S_W)$, where $S_H$ and $S_W$ are *stepsize* along with the height and width dimension respectively (illustrated in Figure 1 (d)). The basic Cycle FC operator can be formulated as below:

$$\text{CycleFC}(\boldsymbol{X})_{i,j,:} = \sum_{c=0}^{C_{in}} \boldsymbol{X}_{i+\delta_i(c),j+\delta_j(c),c} \cdot \boldsymbol{W}_{c,:}^{\text{mlp}} + \boldsymbol{b} \tag{1}$$

where $\boldsymbol{W}^{\text{mlp}} \in \mathbb{R}^{C_{in} \times C_{out}}$ and $\boldsymbol{b} \in \mathbb{R}^{C_{out}}$ are parameters of Cycle FC. $\delta_i(c)$ and $\delta_j(c)$ are the spatial offset of the two axis on the $c^{th}$ channel, which are defined as below:

$$\delta_i(c) = (c \bmod S_H) - 1, \qquad \delta_j(c) = (\lfloor \frac{c}{S_H} \rfloor \bmod S_W) - 1 \tag{2}$$

**Examples.** We provide several examples (Figure 1 (d)-(f)) to illustrate the stepsize. For the sake of visualization convenience, we set the tensor's $W = 1$. Thus, these three examples naturally all have $S_W = 1$. Figure 1 (d) illustrates the offsets along two axis when $S_H = 3$, that is $\delta_j(c) \equiv 0$ and $\delta_i(c) = \{-1, 0, 1, -1, 0, 1, \cdots\}$ when $c = 0, 1, 2, \cdots, 8$. Figure 1 (e) shows that when $S_H = H$, Cycle FC has a global receptive field. Figure 1 (f) shows that when $S_H = 1$, there will be no offset

along either axis and thus Cycle FC degrades to Channel FC (Figure 1 (a)). We also provide a more general case where $W \neq 1$ and $S_H = 3, S_W = 3$ in Figure 7 (Appendix).

The offsets $\delta_i(c)$ and $\delta_j(c)$ enlarge the receptive field of Cycle FC as compared to Channel FC (Figure 1a), which applies weights solely on the same spatial position for all channels. The larger receptive field in return brings improvements on dense prediction tasks like semantic segmentation and object detection as shown in Table 1. Meanwhile, Cycle FC still maintains computational efficiency and flexibility on input resolution. Both the FLOPs and the number of parameters are linear to the spatial scale which are exactly the same as those of Channel FC. In contrast, although Spatial FC has a global receptive field over the whole spatial space, its computational cost is quadratic to the image scale. Besides, it fails to handle inputs with different resolutions.

## 2.2 COMPARISON BETWEEN MULTI-HEAD SELF-ATTENTION (MHSA) AND CYCLE FC

Inspired by Cordonnier et al. (2020), when re-parametried properly, a multi-head self-attention layer with $N_h$ heads can be formulated as below, which is similar to a convolution with kernel size $\sqrt{N_h} \times \sqrt{N_h}$. (Please refer to Appendix C for detailed derivation)

$$\mathrm{MHSA}(\boldsymbol{X})_{i,j,:} = \sum_{h \in \{1,2,\dots,N_h\}} \boldsymbol{X}_{i+\Delta_i(h),j+\Delta_j(h),:} \boldsymbol{W}^{\mathrm{mhsa},h} + \boldsymbol{b} \tag{3}$$

where $\boldsymbol{W}^{\mathrm{mhsa},h} \in \mathbb{R}^{C_{in} \times C_{out}}$ is the parameter matrix for $h^{th}$ head in MHSA. $\boldsymbol{b} \in \mathbb{R}^{C_{out}}$ is the bias vector. $\{\Delta_i(h), \Delta_j(h)\} = \{(0,0), (1,0), (-1,0), \cdots\}$ contains all possible positional shift in convolution with kernel size $\sqrt{N_h} \times \sqrt{N_h}$. Further, we stack all $\boldsymbol{W}^{\mathrm{mhsa},h}$ together and reshape it into $\boldsymbol{W}^{\mathrm{mhsa}} \in \mathbb{R}^{K \times K \times C_{in} \times C_{out}}$. Then a relationship between $\boldsymbol{W}^{\mathrm{mlp}}$ and $\boldsymbol{W}^{\mathrm{mhsa}}$ can be formulated as follow.

$$\boldsymbol{W}_{c,:}^{\mathrm{mlp}} = \boldsymbol{W}_{\delta_i(c)+1,\delta_j(c)+1,c,:}^{\mathrm{mhsa}} \tag{4}$$

equation 4 shows that only the weights of $\boldsymbol{W}^{\mathrm{mhsa}}$ on spatial shift $(\delta_i(c)+1, \delta_j(c)+1)$ are taken into account in $\boldsymbol{W}^{\mathrm{mlp}}$. This indicates that Cycle FC introduce an inductive bias that the weighting matrix in MHSA should be sparse. Thus Cycle FC inherits the large receptive field introduced in MHSA. The receptive field in Cycle FC is enlarged to $(S_H, S_W)$, which enables Cycle FC to tackle with downstream dense prediction tasks better. Meanwhile, with the sparsity inductive bias, Cycle FC maintains computational efficiency in MLP-based methods as compared to convolution and multi-head self-attention. The parameter size in Cycle FC is $C_{in} \times C_{out}$ while $\boldsymbol{W}^{\mathrm{mhsa}} \in \mathbb{R}^{K \times K \times C_{in} \times C_{out}}$.

## 2.3 OVERALL ARCHITECTURE

**Patch Embedding.** Given the raw input image with the size of $H \times W \times 3$, our model first splits it into patches by a patch embedding module (Dosovitskiy et al., 2020). Each patch is then treated as a "token". Specifically, we follow (Fan et al., 2021; Wang et al., 2021a) to adopt an overlapping patch embedding module with the window size 7 and stride 4. These raw patches are further projected to a higher dimension (denoted as $C$) by a linear embedding layer. Therefore, the overall patch embedding module generates the features with the shape of $\frac{H}{4} \times \frac{W}{4} \times C$.

**CycleMLP Block.** Then, we sequentially apply several Cycle FC Bloc blocks. Comparing with the previous MLP blocks (Tolstikhin et al., 2021; Touvron et al., 2021a; Liu et al., 2021a) visualized in Figure 5 (Appendix), the key difference of Cycle FC block is that it utilizes our proposed *Cycle Fully-Connected Layer (Cycle FC)* for spatial projection and advances the models in context aggregation and information communication. Specifically, the Cycle FC block consists of three parallel Cycle FCs, which have stepsizes $S_H \times S_W$ of $1 \times 7$, $7 \times 1$, and $1 \times 1$. This design is inspired by the factorization of convolution (Szegedy et al., 2016) and criss-cross attention (Huang et al., 2019). Then, there is a channel-MLP with two linear layers and a GELU (Hendrycks & Gimpel, 2016) non-linearity in between. A LayerNorm (LN) (Ba et al., 2016) layer is applied before both parallel Cycle FC layers and channel-MLP modules. A residual connection (He et al., 2016) is applied after each module.

| Model | Param | FLOPs | Top-1 |
|---|---|---|---|
| EAMLP-14 | 30M | - | 78.9 |
| EAMLP-19 | 55M | - | 79.4 |
| Mixer-B/16 | 59M | 12.7G | 76.4 |
| Mixer-B/16$^\dagger$ | 59M | 12.7G | 77.3 |
| ResMLP-S12 | 15M | 3.0G | 76.6 |
| ResMLP-S24 | 30M | 6.0G | 79.4 |
| ResMLP-B24 | 116M | 23.0G | 81.0 |
| gMLP-Ti | 6M | 1.4G | 72.3 |
| gMLP-S | 20M | 4.5G | 79.6 |
| gMLP-B | 73M | 15.8G | 81.6 |
| $S^2$-MLP-wide | 71M | 14.0G | 80.0 |
| $S^2$-MLP-deep | 51M | 10.5G | 80.7 |
| ViP-Small/7 | 25M | 6.9G | 81.5 |
| ViP-Medium/7 | 55M | 16.3G | 82.7 |
| ViP-Large/7 | 88M | 24.4G | 83.2 |
| AS-MLP-T | 28M | 4.4G | 81.3 |
| AS-MLP-S | 50M | 8.5G | 83.1 |
| AS-MLP-B | 88M | 15.2G | 83.3 |
| CycleMLP-B1 | 15M | 2.1G | 79.1 |
| CycleMLP-B2 | 27M | 3.9G | 81.6 |
| CycleMLP-B3 | 38M | 6.9G | 82.6 |
| CycleMLP-B4 | 52M | 10.1G | 83.0 |
| CycleMLP-B5 | 76M | 12.3G | 83.1 |
| CycleMLP-T | 28M | 4.4G | 81.3 |
| CycleMLP-S | 50M | 8.5G | 82.9 |
| CycleMLP-B | 88M | 15.2G | **83.4** |

Table 2: ImageNet-1K classification for **MLP-like** models.

| Model | Family | Scale | Param | FLOPs | Top-1 |
|---|---|---|---|---|---|
| ResNet18 | CNN | $224^2$ | 12M | 1.8G | 69.8 |
| EffNet-B3 | CNN | $300^2$ | 12M | 1.8G | 81.6 |
| GFNet-H-Ti | FFT | $224^2$ | 15M | 2.0G | 80.1 |
| CycleMLP-B1 | MLP | $224^2$ | 15M | 2.1G | 78.9 |
| ResNet50 | CNN | $224^2$ | 26M | 4.1G | 78.5 |
| DeiT-S | Trans | $224^2$ | 22M | 4.6G | 79.8 |
| BoT-S1-50 | Hybrid | $224^2$ | 21M | 4.3G | 79.1 |
| PVT-S | Trans | $224^2$ | 25M | 3.8G | 79.8 |
| Swin-T | Trans | $224^2$ | 29M | 4.5G | 81.3 |
| GFNet-H-S | FFT | $224^2$ | 32M | 4.5G | 81.5 |
| CycleMLP-B2 | MLP | $224^2$ | 27M | 3.9G | 81.6 |
| ResNet101 | CNN | $224^2$ | 45M | 7.9G | 79.8 |
| RegNetY-8G | CNN | $224^2$ | 39M | 8.0G | 81.7 |
| BoT-S1-59 | Hybrid | $224^2$ | 34M | 7.3G | 81.7 |
| PVT-M | Trans | $224^2$ | 44M | 6.7G | 81.2 |
| CycleMLP-B3 | MLP | $224^2$ | 38M | 6.9G | 82.4 |
| GFNet-H-B | FFT | $224^2$ | 54M | 8.4G | 82.9 |
| Swin-S | Trans | $224^2$ | 50M | 8.7G | 83.0 |
| PVT-L | Trans | $224^2$ | 61M | 9.8G | 81.7 |
| CycleMLP-S | MLP | $224^2$ | 50M | 8.5G | 82.9 |
| ViT-B/16 | Trans | $384^2$ | 86M | 55.4G | 77.9 |
| DeiT-B | Trans | $224^2$ | 86M | 17.5G | 81.8 |
| DeiT-B | Trans | $384^2$ | 86M | 55.4G | 83.1 |
| Swin-B | Trans | $224^2$ | 88M | 15.4G | 83.3 |
| CycleMLP-B | MLP | $224^2$ | 88M | 15.2G | 83.4 |

Table 3: **Comparison with SOTA models on ImageNet-1K without extra data.**

**Stage.** The blocks with the same architecture are stacked to form one *Stage* (He et al., 2016). The number of tokens (feature scale) is maintained within each stage. At each stage *transition*, the channel capacity of the processed tokens is expanded while the number of tokens is reduced. This strategy effectively reduces the spatial resolution complexity. Overall, each of our model variants has four stages, and the output feature at the last stage has a shape of $\frac{H}{32} \times \frac{W}{32} \times C_4$. These stage settings are widely utilized in both CNN (Simonyan & Zisserman, 2014; He et al., 2016) and Transformer (Wang et al., 2021b; Liu et al., 2021b) models. Therefore, CycleMLP can conveniently serve as a general-purpose visual backbone and a generic replacement for existing backbones.

**Model Variants.** The design principle of the model's macro structure is mainly inspired by the philosophy of hierarchical Transformer (Wang et al., 2021b; Liu et al., 2021b) models, which reduce the number of tokens at the transition layers as the network goes deeper and increase the channel dimension. In this way, we can build a hierarchical architecture that is critical for dense prediction tasks (Lin et al., 2014; Zhou et al., 2017). Specifically, we build two model zoos following two widely used Transformer architectures, PVT (Wang et al., 2021b) and Swin (Liu et al., 2021b). Models in PVT-style are named from CycleMLP-B1 to CycleMLP-B5 and in Swin-Style are named as CycleMLP-T, -S, and -B, which represent models in *tiny, small*, and *base* sizes. These models are built by adapting several architecture-related hyper-parameters, including $S_i$, $C_i$, $E_i$, and $L_i$ which represent the stride of the transition, the token channel dimension, the number of blocks, and the expansion ratio respectively at Stage $i$. Detailed configurations of these models are in Table 11 (Appendix).

## 3 EXPERIMENTS

In this section, we first examine CycleMLP by conducting experiments on ImageNet-1K (Deng et al., 2009) image classification. Then, we present a bunch of baseline models achieved by CycleMLP in dense prediction tasks, *i.e.,* COCO (Lin et al., 2014) object detection, instance segmentation, and ADE20K (Zhou et al., 2017) semantic segmentation.

| $1 \times 7$ | $7 \times 1$ | $1 \times 1$ | Params | FLOPs | Top-1 Acc |
|---|---|---|---|---|---|
| ✓ | | ✓ | | | 80.5 |
| | ✓ | ✓ | 24.5M | 3.6G | 80.4 |
| ✓ | ✓ | | | | 81.3 |
| ✓✓ | | ✓ | 26.8M | 3.9G | 80.6 |
| | ✓✓ | ✓ | | | 80.5 |
| ✓ | ✓ | ✓ | 26.8M | 3.9G | 81.6 |

Table 4: **Ablation on three parallel branches.** We adopt CycleMLP-B2 variant for this ablation study. Double check marks (✓✓) denote two same branches.

| Stepsize | ImgNet Top-1 | ADE20K mIoU |
|---|---|---|
| 3 | 81.6 | 42.4 |
| 5 | 81.6 (+0.0) | 43.2 (+0.8) |
| 7 | 81.6 (+0.0) | **43.9** (+1.5) |
| 9 | 81.5 (−0.1) | 43.2 (+0.8) |

Table 5: *Stepsize* **ablation:** CycleMLP achieves the highest mIoU on ADE20K when stepsize is 7. However, the stepsize has negligible influence on the ImageNet classification.

## 3.1 IMAGENET-1K CLASSIFICATION

The experimental settings for ImageNet classification are mostly from DeiT (Touvron et al., 2020), Swin (Liu et al., 2021b). The detailed experimental settings for ImageNet classification can be found in Appendix E.1.

**Comparison with MLP-like Models.** We first compare CycleMLP with existing MLP-like models and the results are summarized in Table 2 and Figure 2. The accuracy-FLOPs tradeoff of CycleMLP consistently outperforms existing MLP-like models (Tolstikhin et al., 2021; Touvron et al., 2021a; Liu et al., 2021a; Guo et al., 2021; Yu et al., 2021; Hou et al., 2021) under a wide range of FLOPs, which we attribute to the effectiveness of our Cycle FC. Specifically, compared with one of the pioneering MLP work, *i.e.,* gMLP (Liu et al., 2021a), CycleMLP-B2 achieves the same top-1 accuracy (81.6%) as gMLP-B while reducing more than $3\times$ FLOPs (3.9G for CycleMLP-B2 and 15.8G for gMLP-B). Furthermore, compared with existing SOTA MLP-like model, *i.e.,* ViP (Hou et al., 2021), our model CycleMLP-B utilizes less FLOPs (15.2G) than ViP-Large/7 (24.4G, the largest one of ViP family) while achiving higher top-1 accuracy.

It is noted that all previous MLP-like models listed in Table 2 do not conduct experiments on dense prediction tasks due to the incapability of dealing with variable input scales, which is discussed in Sec. 1. However, CycleMLP solved this issue by adopting Cycle FC. The experimental results on dense prediction tasks are presented in Sec. 3.3 and Sec. 3.4.

**Comparison with SOTA Models.** Table 3 further compares CycleMLP with previous state-of-the-art CNN, Transformer and Hybrid architectures. It is interesting to see that CycleMLP models achieve comparable performance to Swin Transformer (Liu et al., 2021b), which is the state-of-the-art Transformer-based model. Specifically, CycleMLP-B achieves slightly better top-1 accuracy (83.4%) than Swin-B (83.3%) with similar parameters and FLOPs. GFNet (Rao et al., 2021) utilizes the fast Fourier transform (FFT) (Cooley & Tukey, 1965) to learn spatial information and achieves similar performance as CycleMLP on ImageNet-1K classification. However, the architecture of GFNet is correlated with the input resolution, and extra operation (parameter interpolation) is required when input scale changes, which may hurt the performance of dense predictions. We will thoroughly compare CycleMLP with GFNet in Sec. 3.4 on ADE20K.

## 3.2 ABLATION STUDY

In this subsection, we conduct extensive ablation studies to analyze each component of our design. Unless otherwise stated, We adopt CycleMLP-B2 instantiation in this subsection.

**Cycle Fully-Connected Layer.** To demonstrate the advantage of the Cycle FC, we compare CycleMLP-B2 with two other baseline models equipped with channel FC and Spatial FC as spatial context aggregation operators, respectively. The differences of these operators are visualized in Figure 1, and the comparison results are shown in Table 1. CycleMLP-B2 outperforms the counterparts built on both Spatial and Channel FC for ImageNet classification, COCO object detection, instance segmentation, and ADE20K semantic segmentation. The results validate that Cycle FC is capable of serving as a general-purpose, plug-and-play operator for spatial information communication and context aggregation.

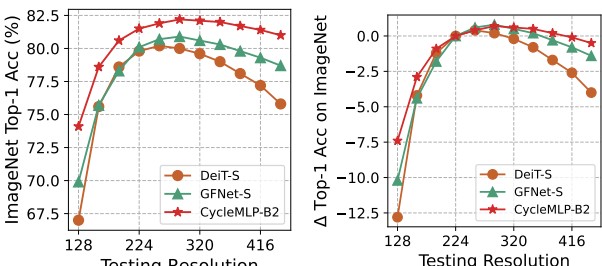

Figure 3: **Resolution adaptability.** All models are trained on 224×224 and evaluated on various resolutions without fine-tuning. **Left:** Absolute top-1 accuracy; **Right:** Accuracy difference relative to that tested on 224×224. The superiority of CycleMLP's robustness becomes more significant when scale varies to a greater extent.

Table 4 further details the ablation study on the structure of CycleMLP block. It is observed that the top-1 accuracy drops significantly after removing one of the three parallel branches, especially when discarding the 1×7 or 7×1 branch. To eliminate the probability that the fewer parameters and FLOPs cause the performance drop, we further use two same branches (denoted as "✓✓" in Table 4) and one 1×1 branch to align the parameters and FLOPs. The accuracy still drops relative to CycleMLP, which further demonstrates the necessity of these three unique branches.

**Resolution adaptability.** One remarkable advantage of CycleMLP is that it can take arbitrary-resolution images as input without any modification. On the contrary, GFNet (Rao et al., 2021) needs to interpolate the learnable parameters on the fly when the input scale is different from the one for training. We compare the resolution adaptability by directly evaluating models at a broad spectrum of resolutions using the weight pre-trained on 224×224, without fine-tuning. Figure 3 (left) shows that the absolute Top-1 accuracy on ImagNet and Figure 3 (right) shows the accuracy differences between one specific resolution and the resolution of 224×224. Compared with DeiT and GFNet, CycleMLP is more robust when resolution varies. In particular, at the 128×128, CycleMLP saves more than 2 points drop compared to GFNet. Furthermore, at higher resolution, the performance drop of CycleMLP is less than GFNet. Note that the superiority of CycleMLP becomes more significant when the resolution changes to a greater extent.

## 3.3 OBJECT DETECTION AND INSTANCE SEGMENTATION

**Settings.** We conduct object detection and instance segmentation experiments on COCO (Lin et al., 2014) dataset. We first follow the experimental settings of PVT (Wang et al., 2021b), which are introduced in Appendix. E.2. The corresponding results are presented in Table 6. Then, in order to compare fairly with Swin Transformer, which adopts a different experimental recipe with PVT, we further follow the experimental settings of Swin with our CycleMLP-S model and the results are presented in Table 7.

| Backbone | RetinaNet 1× | | | | | | | Mask R-CNN 1× | | | | | |
|---|---|---|---|---|---|---|---|---|---|---|---|---|---|
| | Param | AP | $AP_{50}$ | $AP_{75}$ | $AP_S$ | $AP_M$ | $AP_L$ | Param | $AP^b$ | $AP_{50}^b$ | $AP_{75}^b$ | $AP^m$ | $AP_{50}^m$ | $AP_{75}^m$ |
| ResNet18 | 21.3M | 31.8 | 49.6 | 33.6 | 16.3 | 34.3 | 43.2 | 31.2M | 34.0 | 54.0 | 36.7 | 31.2 | 51.0 | 32.7 |
| PVT-Tiny | 23.0M | 36.7 | 56.9 | 38.9 | 22.6 | 38.8 | 50.0 | 32.9M | 36.7 | 59.2 | 39.3 | 35.1 | 56.7 | 37.3 |
| CycleMLP-B1 | 24.9M | 38.1 | 58.7 | 40.1 | 21.9 | 41.9 | 50.4 | 34.8M | 39.8 | 61.7 | 43.3 | 37.0 | 58.8 | 39.7 |
| ResNet50 | 37.7M | 36.3 | 55.3 | 38.6 | 19.3 | 40.0 | 48.8 | 44.2M | 38.0 | 58.6 | 41.4 | 34.4 | 55.1 | 36.7 |
| PVT-Small | 34.2M | 40.4 | 61.3 | 43.0 | 25.0 | 42.9 | 55.7 | 44.1M | 40.4 | 62.9 | 43.8 | 37.8 | 60.1 | 40.3 |
| CycleMLP-B2 | 36.5M | 40.6 | 61.4 | 43.2 | 22.9 | 44.4 | 54.5 | 46.5M | 42.1 | 64.0 | 45.7 | 38.9 | 61.2 | 41.8 |
| ResNet101 | 56.7M | 38.5 | 57.8 | 41.2 | 21.4 | 42.6 | 51.1 | 63.2M | 40.4 | 61.1 | 44.2 | 36.4 | 57.7 | 38.8 |
| ResNeXt101-32x4d | 56.4M | 39.9 | 59.6 | 42.7 | 22.3 | 44.2 | 52.5 | 62.8M | 41.9 | 62.5 | 45.9 | 37.5 | 59.4 | 40.2 |
| PVT-Medium | 53.9M | 41.9 | 63.1 | 44.3 | 25.0 | 44.9 | 57.6 | 63.9M | 42.0 | 64.4 | 45.6 | 39.0 | 61.6 | 42.1 |
| CycleMLP-B3 | 48.1M | 42.5 | 63.2 | 45.3 | 25.2 | 45.5 | 56.2 | 58.0M | 43.4 | 65.0 | 47.7 | 39.5 | 62.0 | 42.4 |
| PVT-Large | 71.1M | 42.6 | 63.7 | 45.4 | 25.8 | 46.0 | 58.4 | 81.0M | 42.9 | 65.0 | 46.6 | 39.5 | 61.9 | 42.5 |
| CycleMLP-B4 | 61.5M | 43.2 | 63.9 | 46.2 | 26.6 | 46.5 | 57.4 | 71.5M | 44.1 | 65.7 | 48.1 | 40.2 | 62.7 | 43.5 |
| ResNeXt101-64x4d | 95.5M | 41.0 | 60.9 | 44.0 | 23.9 | 45.2 | 54.0 | 101.9M | 42.8 | 63.8 | 47.3 | 38.4 | 60.6 | 41.3 |
| CycleMLP-B5 | 85.9M | 42.7 | 63.3 | 45.3 | 24.1 | 46.3 | 57.4 | 95.3M | 44.1 | 65.5 | 48.4 | 40.1 | 62.8 | 43.0 |

Table 6: **Object detection and instance segmentation on COCO `val2017` (Lin et al., 2014).** We compare CycleMLP with various backbones including ResNet (He et al., 2016), ResNeXt (Xie et al., 2017) and PVT (Wang et al., 2021b).

| Backbone | $AP^b$ | $AP^b_{50}$ | $AP^b_{75}$ | $AP^m$ | $AP^m_{50}$ | $AP^m_{75}$ | Params | FLOPs |
|---|---|---|---|---|---|---|---|---|
| ResNet50 (He et al., 2016) | 41.0 | 61.7 | 44.9 | 37.1 | 58.4 | 40.1 | 44M | 260G |
| PVT-Small (Wang et al., 2021b) | 43.0 | 65.3 | 46.9 | 39.9 | 62.5 | 42.8 | 44M | 245G |
| Swin-T (Liu et al., 2021b) | 46.0 | 68.2 | 50.2 | 41.6 | **65.1** | 44.8 | 48M | 264G |
| CycleMLP-T (**ours**) | **46.4** | **68.1** | **51.1** | **41.8** | 64.9 | **45.1** | 48M | 260G |

Table 7: The instance segmentation results of different backbones on the COCO val2017 dataset. Mask R-CNN frameworks are employed.

**Results.** Firstly, as shown in Table 6, CycleMLP-based RetinaNet consistently surpasses the CNN-based ResNet (He et al., 2016), ResNeXt (Xie et al., 2017) and Transformer-based PVT (Wang et al., 2021b) under similar parameter constraints, indicating that CycleMLP can serve as an excellent general-purpose backbone. Furthermore, using Mask R-CNN (He et al., 2017) for instance segmentation also demonstrates similar comparison results. Furthermore, from Table 7, the CycleMLP can achieve a slightly better performance than Swin Transformer.

## 3.4 SEMANTIC SEGMENTATION

**Settings.** We conduct semantic segmentation experiments on ADE20K (Zhou et al., 2017) dataset and present the detailed settings in Appendix E.3. Table 8 and Table 9 show the experimental results using training recipes from PVT and Swin respectively.

| Backbone | Semantic FPN | |
|---|---|---|
| | Param | mIoU (%) |
| ResNet18 (He et al., 2016) | 15.5M | 32.9 |
| PVT-Tiny (Wang et al., 2021b) | 17.0M | 35.7 |
| CycleMLP-B1 (ours) | 18.9M | **40.8** |
| ResNet50 (He et al., 2016) | 28.5M | 36.7 |
| PVT-Small (Wang et al., 2021b) | 28.2M | 39.8 |
| Swin-T† (Liu et al., 2021b) | 31.9M | 41.5 |
| GFNet-Tiny (Rao et al., 2021) | 26.6M | 41.0 |
| CycleMLP-B2 (ours) | 30.6M | **43.4** |
| ResNet101 (He et al., 2016) | 47.5M | 38.8 |
| ResNeXt101-32x4d (Xie et al., 2017) | 47.1M | 39.7 |
| PVT-Medium (Wang et al., 2021b) | 48.0M | 41.6 |
| GFNet-Small (Rao et al., 2021) | 47.5M | 42.5 |
| CycleMLP-B3 (ours) | 42.1M | **44.3** |
| PVT-Large (Wang et al., 2021b) | 65.1M | 42.1 |
| Swin-S† (Liu et al., 2021b) | 53.2M | **45.2** |
| CycleMLP-B4 (ours) | 55.6M | 45.1 |
| GFNet-Base (Rao et al., 2021) | 74.7M | 44.8 |
| ResNeXt101-64x4d (Xie et al., 2017) | 86.4M | 40.2 |
| CycleMLP-B5 (ours) | 79.4M | **45.5** |

Table 8: **Semantic segmentation on ADE20K (Zhou et al., 2017) val.** All models are equipped with Semantic FPN (Kirillov et al., 2019). † Results are from GFNet (Rao et al., 2021).

(a) Swin

(b) CycleMLP

Figure 4: **Effective Receptive Field (ERF).** We visualize the ERFs of the last stage for both Swin (Liu et al., 2021b) and CycleMLP. Best viewed with zoom in.

**Results.** As shown in Table 8, CycleMLP outperforms ResNet (He et al., 2016) and PVT (Wang et al., 2021b) significantly with similar parameters. Moreover, compared to the state-of-the-art Transformer-based backbone, Swin Transformer (Liu et al., 2021b), CycleMLP can obtain comparable or even better performance. Specifically, CycleMLP-B2 surpasses Swin-T by 0.9 mIoU with slightly less parameters (30.6M *v.s.* 31.9M).

Although GFNet (Rao et al., 2021) achieves similar performance as CycleMLP on ImageNet classification, CycleMLP notably outperforms GFNet on ADE20K semantic segmentation where input scale varies. We attribute the superiority of CycleMLP under a scale-variable scenario to the capability of dealing with arbitrary scales. On the contrary, GFNet (Rao et al., 2021) requires additional

| Method | Backbone | val MS mIoU | Params | FLOPs |
|---|---|---|---|---|
| UperNet (Xiao et al., 2018) | Swin-T (Liu et al., 2021b) | 45.8 | 60M | 945G |
| | AS-MLP-T (Lian et al., 2021) | 46.5 | 60M | 937G |
| | CycleMLP-T (ours) | **47.1** | 60M | 937G |
| UperNet (Xiao et al., 2018) | Swin-S (Liu et al., 2021b) | 49.5 | 81M | 1038G |
| | AS-MLP-S (Lian et al., 2021) | 49.2 | 81M | 1024G |
| | CycleMLP-S (ours) | **49.6** | 81M | 1024G |
| UperNet (Xiao et al., 2018) | Swin-B (Liu et al., 2021b) | 49.7 | 121M | 1188G |
| | AS-MLP-B(Lian et al., 2021) | 49.5 | 121M | 1166G |
| | CycleMLP-B (ours) | **49.7** | 121M | 1166G |

Table 9: The semantic segmentation results of different backbones on the ADE20K validation set.

heuristic operation (weight interpolation) when the input scale varies, which may hurt the performance.

Moreover, we also visualized the receptive field following (Xie et al., 2021), and the results are visualized in Figure 4, which demonstrate that our CycleMLP has a larger effective receptive field than Swin.

## 3.5 ROBUSTNESS

| Network | mCE↓ | Noise | | | Blur | | | | Weather | | | | Digital | | | |
|---|---|---|---|---|---|---|---|---|---|---|---|---|---|---|---|---|
| | | Gauss | Shot | Impulse | Defocus | Glass | Motion | Zoom | Snow | Frost | Fog | Bright | Contrast | Elastic | Pixel | JPEG |
| ResNet-50 | 76.7 | 79.8 | 81.6 | 82.6 | 74.7 | 88.6 | 78.0 | 79.9 | 77.8 | 74.8 | 66.1 | 56.6 | 71.4 | 84.7 | 76.9 | 76.8 |
| DeiT-S | 54.6 | 46.3 | 47.7 | 46.4 | 61.6 | **71.9** | 57.9 | 71.9 | 49.9 | **46.2** | **46.0** | 44.9 | 42.3 | **66.6** | 59.1 | 60.4 |
| Swin-S | 62.0 | 52.2 | 53.7 | 53.6 | 67.9 | 78.6 | 64.1 | 75.3 | 55.8 | 52.8 | 51.3 | 48.1 | 45.1 | 75.7 | 76.3 | 79.1 |
| MLP-Mixer | 78.8 | 80.9 | 82.6 | 84.2 | 86.9 | 92.1 | 79.1 | 93.6 | 78.3 | 67.4 | 64.6 | 59.5 | 57.1 | 90.5 | 72.7 | 92.2 |
| ResMLP-12 | 66.0 | 57.6 | 58.2 | 57.8 | 72.6 | 83.2 | 67.9 | 76.5 | 61.4 | 57.8 | 63.8 | 53.9 | 52.1 | 78.3 | 72.9 | 75.3 |
| gMLP-S | 64.0 | 52.1 | 53.2 | 52.5 | 73.1 | 77.6 | 64.6 | 79.9 | 77.7 | 78.8 | 54.3 | 55.3 | 43.6 | 70.6 | 58.6 | 67.5 |
| CycleMLP-S | **53.7** | **42.1** | **43.4** | **43.2** | **61.5** | 76.7 | **56.0** | **66.4** | **51.5** | 47.2 | 50.8 | **41.2** | **39.5** | 72.3 | **57.5** | 56.1 |

Table 10: **Robustness on ImageNet-C (Hendrycks & Dietterich, 2019).** The mean corruption error (**mCE**) normalized by AlexNet (Krizhevsky et al., 2012) errors is used as the robustness metric. The lower, the better.

We further conduct experiments on ImageNet-C (Hendrycks & Gimpel, 2016) to analyze the robustness ability of the CycleMLP, following (Mao et al., 2021) and results are presented in Table 10. Compared with both Transformers (*e.g.* DeiT and Swin) and existing MLP models (*e.g.* MLP-Mixer, ResMLP, gMLP), CycleMLP achieves a stronger robustness ability.

## 4 CONCLUSION

We present a versatile MLP-like architecture, CycleMLP, in this work. CycleMLP is built upon the Cycle Fully-Connected Layer (Cycle FC), which is capable of dealing with variable input scales and can serve as a generic, plug-and-play replacement of vanilla FC layers. Experimental results demonstrate that CycleMLP outperforms existing MLP-like models on ImageNet classification and achieves promising performance on dense prediction tasks, *i.e.,* object detection, instance segmentation and semantic segmentation. This work indicates that an attention-free architecture can also serve as a general vision backbone.

**Acknowledgment.** Ping Luo is supported by the General Research Fund of HK No.27208720, No.17212120, and the HKU-TCL Joint Research Center for Artificial Intelligence.

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

## A  LITERATURE ON VISION MODEL

**CNN-based Models.**  Originally introduced over twenty years ago (LeCun et al., 1989), convolutional neural networks (CNN) have been widely adopted since the success of the AlexNet (Krizhevsky et al., 2012) which outperformed prevailing approaches based on hand-crafted image features. There have been several attempts made to improve the design of CNN-based models. VGG (Simonyan & Zisserman, 2014) demonstrated a state-of-the-art performance on ImageNet via deploying small $(3 \times 3)$ convolution kernels to all layers. He *et al.*introduced skip-connections in ResNets (He et al., 2016), enabling a model variant with more than 1000 layers. DenseNet (Huang et al., 2017) connected each layer to every other layer in a feed-forward fashion, strengthening feature propagation and reducing the number of parameters. In parallel with these architecture design works, some other works also made significant contributions to the popularity of CNNs, including normalization (Ioffe & Szegedy, 2015; Ba et al., 2016), data augmentation (Cubuk et al., 2020; Yun et al., 2019; Zhang et al., 2017), etc.

**Transformer-based Models.** Transformers were first proposed by Vaswani *et al.*for machine translation and have since become the dominant choice in many NLP tasks (Devlin et al., 2018; Wang et al., 2018; Yang et al., 2019; Brown et al., 2020). Recently, transformer have also led to a series of breakthroughs in computer vision community since the invention of ViT (Dosovitskiy et al., 2020), and have been working as a *de facto* standard for various tasks, *e.g.,* image classification (Dosovitskiy et al., 2020; Touvron et al., 2020; Yuan et al., 2021), detection and segmentation (Wang et al., 2021b; Liu et al., 2021b; Zheng et al., 2021; Xie et al., 2021), video recognition (Wang et al., 2021c; Bertasius et al., 2021; Arnab et al., 2021; Fan et al., 2021) and so on. Moreover, there has also been lots of interest in adopting transformer to cross aggregate multiple modality information (Radford et al., 2021; Gabeur et al., 2020; Dzabraev et al., 2021). Furthermore, combining CNNs and transformers is also explored in (Srinivas et al., 2021; Li et al., 2021; Wu et al., 2021; Touvron et al., 2021b).

**MLP-based Models.** MLP-based models (Tolstikhin et al., 2021; Touvron et al., 2021a; Liu et al., 2021a) differ from the above discussed CNN- and Transformer-based models because they resort to neither convolution nor self-attention layers. Instead, they use MLP layers over feature patches on spatial dimensions to aggregate the spatial context. These MLP-based models share similar macro structures but differ from each other in the detailed design of the micro block. In addition, MLP-based models provide more efficient computation than transformer-based models since they do not need to calculate affinity matrix using key-query multiplication. Concurrent to our work, $S^2$-MLP (Yu et al., 2021) utilizes a spatial-shift operation for spatial information communication. The similar aspect between our work and $S^2$-MLP lies in that we all conduct MLP operations along the channel dimension. However, our Cycle FC is different from $S^2$-MLP in: (1) $S^2$-MLP achieves communications between patches by splitting feature maps along channel dimension into several groups and shifting different groups in different directions. It introduces extra splitting and shifting operations on the feature map. On the contrary, we propose a novel operator-Cycle Fully-Connected Layer-for spatial context aggregation. It does not modify the feature map and is formulated as a generic, plug-and-play MLP unit that can be used as a direct replacement of vanilla without any adjustments. (2) We design a pyramid structure for and conduct extensive experiments on classification, object detection, instance segmentation, and semantic segmentation. However, the output feature map of $S^2$-MLP has only one single scale in low resolution, which is unsuitable for dense prediction tasks. Only ImageNet classification is evaluated on $S^2$-MLP. We compared Cycle FC with $S^2$-MLP in details in the Section 3.

## B  COMPARISON OF MLP BLOCKS

We summary MLP blocks proposed by recent MLP-related works in Figure 5. We notice that existing MLP blocks, *i.e.,* MLP-Mixer, ResMLP and gMLP share similar method of **Spatial Proj**: Transpose → Fully-Connected over spatial dimension → Transpose back. These models can not cope with variable image scales as the FC layers in Spatial Proj are configured by the `seq_len`.

The blocks used for building CycleMLP consist of our proposed novel Cycle FC, whose configuration has nothing to do with image scales and can naturally deal with dynamic image scales.

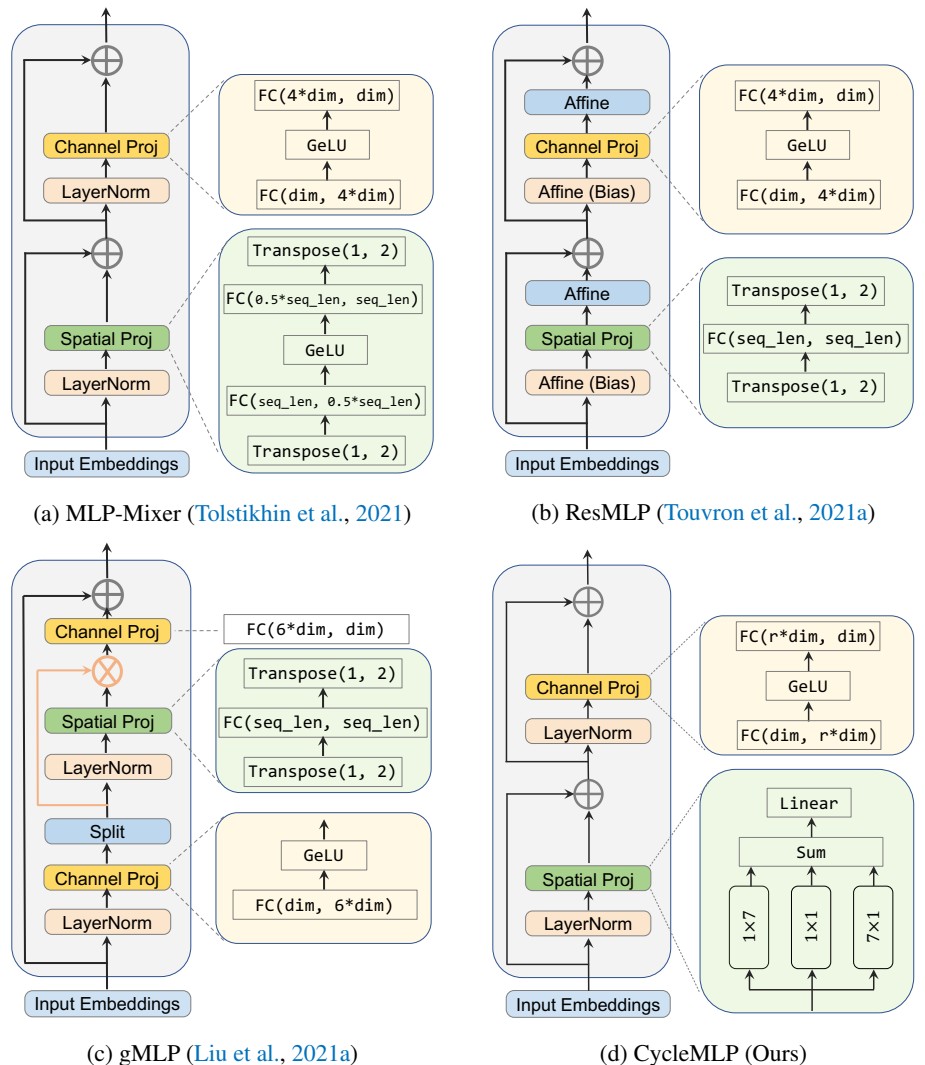

Figure 5: Comparison of MLP blocks in details.

## C   FROM MULTI-HEAD SELF-ATTENTION TO CONVOLUTION

In this section, we provide details in how MHSA can be transferred into a convolution-like operator in equation 3. To start with, the a MHSA layer can be formulated as below:

$$\text{MHSA}(\boldsymbol{X}) = \underset{h \in \{1, \dots, N_h\}}{\text{concat}} [\text{SA}_h(\boldsymbol{X})] \boldsymbol{W}^{out} + \boldsymbol{b} \tag{5}$$

where $\boldsymbol{W}^{out} \in \mathbb{R}^{(N_h C_{out}) \times C'_{out}}$ and $\boldsymbol{b} \in \mathbb{R}^{C'_{out}}$ are parameters for the final linear projection. $\text{SA}_h$ is the $h^{th}$ self-attention module. Then we reshape $\boldsymbol{X}$ into $\boldsymbol{X} \in \mathbb{R}^{HW \times C_{in}}$ and let $T = H \times W$, which indicates that there are $T$ tokens in $\boldsymbol{X}$. $\text{SA}_h$ can be defined as follow:

$$\begin{aligned} \text{SA}(\boldsymbol{X})_{t,:} &= \text{softmax}(\boldsymbol{A}_{t,:})\boldsymbol{V} + \boldsymbol{b} \\ \boldsymbol{A} &= (\boldsymbol{Q} + \boldsymbol{P})(\boldsymbol{K} + \boldsymbol{P})^{\intercal} \end{aligned} \tag{6}$$

where $\boldsymbol{V} = \boldsymbol{X}\boldsymbol{W}^{\text{val}}, \boldsymbol{Q} = \boldsymbol{X}\boldsymbol{W}^{\text{qry}}, \boldsymbol{K} = \boldsymbol{X}\boldsymbol{W}^{\text{key}}$ are respectively the value, query and key matrix with learnable matrices $\boldsymbol{W}^v \in \mathbb{R}^{C_{in} \times C_{out}}, \boldsymbol{W}^q \in \mathbb{R}^{C_{in} \times C_k}, \boldsymbol{W}^k \in \mathbb{R}^{C_{in} \times C_k}$. $\boldsymbol{P} \in \mathbb{R}^{T \times C_{in}}$ is the

positional embedding matrix containing positional information for every input token, which can be replaced by the output of any function $f_P$ that encodes the position of tokens. And $\boldsymbol{A} \in \mathbb{R}^{T \times T}$ is the attention matrix where each element $\boldsymbol{A}_{i,j}$ is the attention score between the $i^{th}$ and $j^{th}$ token in $\boldsymbol{X}$. With absolute positional encoding, the second line in equation 6 can be expanded as (Cordonnier et al., 2020):

$$
\begin{aligned}
\boldsymbol{A}_{q,k} =& (\boldsymbol{X}_{q,:} + \boldsymbol{P}_{q,:})\boldsymbol{W}^{\text{qry}}((\boldsymbol{X}_{k,:} + \boldsymbol{P}_{k,:})\boldsymbol{W}^{\text{key}})^{\intercal} \\
=& \boldsymbol{X}_{q,:}\boldsymbol{W}^{\text{qry}}(\boldsymbol{X}_{k,:}\boldsymbol{W}^{\text{key}})^{\intercal} + \boldsymbol{X}_{q,:}\boldsymbol{W}^{\text{qry}}(\boldsymbol{P}_{k,:}\boldsymbol{W}^{\text{key}})^{\intercal} + \boldsymbol{P}_{q,:}\boldsymbol{W}^{\text{qry}}(\boldsymbol{X}_{k,:}\boldsymbol{W}^{\text{key}})^{\intercal} + \boldsymbol{P}_{q,:}\boldsymbol{W}^{\text{qry}}(\boldsymbol{P}_{k,:}\boldsymbol{W}^{\text{key}})^{\intercal}
\end{aligned}
\tag{7}
$$

When we apply relative positional encoding scheme in (Dai et al., 2019), $\boldsymbol{A}$ is re-parametried into:

$$
\boldsymbol{A}_{q,k} = \boldsymbol{X}_{q,:}\boldsymbol{W}^{\text{qry}}(\boldsymbol{X}_{k,:}\boldsymbol{W}^{\text{key}})^{\intercal} + \boldsymbol{X}_{q,:}\boldsymbol{W}^{\text{qry}}(\boldsymbol{r}_{\delta_{q,k}}\hat{\boldsymbol{W}}^{\text{key}})^{\intercal} + \boldsymbol{u}(\boldsymbol{X}_{k,:}\boldsymbol{W}^{\text{key}})^{\intercal} + \boldsymbol{v}(\boldsymbol{r}_{\delta_{q,k}}\hat{\boldsymbol{W}}^{\text{key}})^{\intercal} \tag{8}
$$

where $\boldsymbol{r}_{\delta_{q,k}}$ is a positional encoding for relative distance $\delta_{q,k} = (\delta_1, \delta_2)$ between token $q$ and $k$ in $\boldsymbol{X}$. $\hat{\boldsymbol{W}}^{key}$ is introduced to only pertain to the positional encoding $\boldsymbol{r}_{q,k}$. $\boldsymbol{u}$ and $\boldsymbol{v}$ are learnable parameter vectors that replace the original $\boldsymbol{P}_{q,:}\boldsymbol{W}^{\text{qry}}$ term, which implies that the attention bias remains the same regardless of the absolution positions of the query. If we set $\boldsymbol{W}^{qry} = \boldsymbol{W}^{key} = 0$ and $\hat{\boldsymbol{W}}^{key} = \boldsymbol{I}$, the first three terms in equation 8 vanish and $\boldsymbol{A}_{q,k} = \boldsymbol{v}\boldsymbol{r}_{q,k}^{\intercal}$. We set $\{\Delta_i(h), \Delta_j(h)\} = \{(0,0), (1,0), (-1,0), \cdots\}$ contains all possible positional shift in convolution with kernel size $\sqrt{N_h} \times \sqrt{N_h}$. For each head $h$, let $\boldsymbol{r}_{q,k} = (\|\delta_{q,k}\|^2, \delta_1, \delta_2)$ and $v^h = -\alpha^h(1, -2\Delta_i(h), -2\Delta_j(h))$, each softmax attention matrix becomes:

$$
\text{softmax}(\boldsymbol{A}^h)_{q,k} = \begin{cases} 1 & if\ \delta_{q,k} = (\Delta_i(h), \Delta_j(h)) \\ 0 & otherwise \end{cases} \tag{9}
$$

Substitute $\text{softmax}(\boldsymbol{A}^h)$ into equation 5 and we get

$$
\text{MHSA}(\boldsymbol{X})_{i,j,:} = \sum_{h \in \{1,2,\dots,N_h\}} \boldsymbol{X}_{i+\Delta_i(h), j+\Delta_j(h),:}\boldsymbol{W}^{\text{mhsa},h} + \boldsymbol{b} \tag{10}
$$

## D ARCHITECTURE VARIANTS

In order to conduct fair and convenient comparison, we build two model zoos: the one is in PVT-Style (named as CycleMLP-B1 to -B5) and the other in Swin-Style (named as CycleMLP-T, -S and -B). These models are scaled up by adapting several architecture-related hyper-parameters, including $S_i$, $C_i$, $E_i$ and $L_i$ which represent the stride of the transition, the token channel dimension, the number of blocks and the expansion ratio respectively at Stage $i$. Detailed configurations of these models are in Table 11.

## E EXPERIMENTAL SETUPS

### E.1 IMAGENET CLASSIFICATION

**Settings.** We train our models on the ImageNet-1K dataset (Deng et al., 2009), which contains 1.2M training images and 50K validation images evenly spreading 1,000 categories. We follow the standard practice in the community by reporting the top-1 accuracy on the validation set. Our code is implemented based on PyTorch (Paszke et al., 2019) framework and heavily relies on the timm (Wightman, 2019) repository. For apple-to-apple comparison, our training strategy is mostly adopted from DeiT (Touvron et al., 2020), which includes RandAugment (Cubuk et al., 2020), Mixup (Zhang et al., 2017), Cutmix (Yun et al., 2019) random erasing (Zhong et al., 2020) and stochastic depth (Huang et al., 2016). The optimizer is AdamW (Loshchilov & Hutter, 2017) with the momentum of 0.9 and weight decay of $5 \times 10^{-2}$ by default. The cosine learning rate schedule is adopted with the initial value of $1 \times 10^{-3}$. All models are trained for 300 epochs on 8 Tesla V100 GPUs with a total batch size of 1024.

| | Output Size | Layer Name | PVT-Style (Wang et al., 2021b) | | | | | Swin-Style (Liu et al., 2021b) | | |
|---|---|---|---|---|---|---|---|---|---|---|
| | | | B1 | B2 | B3 | B4 | B5 | Tiny | Small | Base |
| Stage 1 | $\frac{H}{4} \times \frac{W}{4}$ | Overlapping Patch Embedding | $S_1 = 4$ | | | | | $S_1 = 4$ | | |
| | | | $C_1 = 64$ | | | | $C_1 = 96$ | $C_1 = 96$ | | $C_1 = 128$ |
| | | CycleMLP Block | $E_1 = 4$ $L_1 = 2$ | $E_1 = 4$ $L_1 = 2$ | $E_1 = 8$ $L_1 = 3$ | $E_1 = 8$ $L_1 = 3$ | $E_1 = 4$ $L_1 = 3$ | $E_1 = 4$ $L_1 = 2$ | $E_1 = 4$ $L_1 = 2$ | $E_1 = 4$ $L_1 = 2$ |
| Stage 2 | $\frac{H}{8} \times \frac{W}{8}$ | Overlapping Patch Embedding | $S_2 = 2$ | | | | | $S_2 = 2$ | | |
| | | | $C_2 = 128$ | | | | $C_2 = 192$ | $C_2 = 192$ | | $C_2 = 256$ |
| | | CycleMLP Block | $E_2 = 4$ $L_2 = 2$ | $E_2 = 4$ $L_2 = 3$ | $E_2 = 8$ $L_2 = 4$ | $E_2 = 8$ $L_2 = 8$ | $E_2 = 4$ $L_2 = 4$ | $E_1 = 4$ $L_1 = 2$ | $E_1 = 4$ $L_1 = 2$ | $E_1 = 4$ $L_1 = 2$ |
| Stage 3 | $\frac{H}{16} \times \frac{W}{16}$ | Overlapping Patch Embedding | $S_3 = 2$ | | | | | $S_3 = 2$ | | |
| | | | $C_3 = 320$ | | | | $C_3 = 384$ | $C_3 = 384$ | | $C_3 = 512$ |
| | | CycleMLP Block | $E_3 = 4$ $L_3 = 4$ | $E_3 = 4$ $L_3 = 10$ | $E_3 = 4$ $L_3 = 18$ | $E_3 = 4$ $L_3 = 27$ | $E_3 = 4$ $L_3 = 24$ | $E_1 = 4$ $L_1 = 6$ | $E_1 = 4$ $L_1 = 18$ | $E_1 = 4$ $L_1 = 18$ |
| Stage 4 | $\frac{H}{32} \times \frac{W}{32}$ | Overlapping Patch Embedding | $S_4 = 2$ | | | | | $S_4 = 2$ | | |
| | | | $C_4 = 512$ | | | | $C_4 = 768$ | $C_4 = 768$ | | $C_4 = 1024$ |
| | | CycleMLP Block | $E_4 = 4$ $L_4 = 2$ | $E_4 = 4$ $L_4 = 3$ | $E_4 = 4$ $L_4 = 3$ | $E_4 = 4$ $L_4 = 3$ | $E_4 = 4$ $L_4 = 3$ | $E_1 = 4$ $L_1 = 2$ | $E_1 = 4$ $L_1 = 2$ | $E_1 = 4$ $L_1 = 2$ |
| | Parameters (M) | | 15.2 | 26.8 | 38.4 | 51.8 | 75.7 | 28.3 | 49.6 | 87.7 |
| | FLOPs (G) | | 2.1 | 3.9 | 6.9 | 10.1 | 12.3 | 4.4 | 8.6 | 15.2 |

Table 11: **Instantiations of the CycleMLP with varying complexity.** The $E_i$ and $L_i$ denote the *expand ratio* and *number of repeated layers*. Our design principle is inspired by the philosophy of ResNet (He et al., 2016), where the channel dimension increases while the spatial resolution shrinks with the layer going deeper.

Further kernel optimization for Cycle FC may bring a faster speed but is beyond the scope of this work.

### E.2 COCO INSTANCE SEGMENTATION

We conduct object detection and instance segmentation experiments on COCO (Lin et al., 2014) dataset, which contains 118K and 5K images for `train` and `validation` splits. We adopt the `mmdetection` (Chen et al., 2019) toolbox for all experiments in this subsection. To evaluate the our CycleMLP backbones, we adopt two widely used detectors, *i.e.,* RetinaNet (Lin et al., 2017) and Mask R-CNN (He et al., 2017). All backbones are initialized with ImageNet pre-trained weights and other newly added layers are initialized via Xavier (Glorot & Bengio, 2010). We use the AdamW (Loshchilov & Hutter, 2017) optimizer with the initial learning rate of $1 \times 10^{-4}$. All models are trained on 8 Tesla V100 GPUs with a total batch size of 16 for 12 epochs (*i.e.,* $1 \times$ training scheduler). The input images are resized to the shorted side of 800 pixels and the longer side does not exceed 1333 pixels during training. We do not use the multi-scale (Carion et al., 2020; Zhu et al., 2020; Sun et al., 2021) training strategy. In the testing stage, the shorter side of input images is resized to 800 pixels while no constraint on the longer side.

### E.3 ADE20K SEMANTIC SEGMENTATION

We conduct semantic segmentation experiments on ADE20K (Zhou et al., 2017) dataset, which covers a broad range of 150 semantic categories. ADE20K contains 20K training, 2K validation and 3K testing images. We adopt the `mmsegmenation` (Contributors, 2020) toolbox as our codebase in this subsection. The experimental settings mostly follow PVT (Wang et al., 2021b), which trains models for 40K iterations on 8 Tesla V100 GPUs with 4 samples per GPU. The backbone is initialized with the pre-trained weights on ImageNet. All models are optimized by AdamW (Loshchilov & Hutter, 2017). The initial learning rate is configured as $2 \times 10^{-4}$ with the polynomial decay parameter of 0.9. Input images are randomly resized and cropped to $512 \times 512$ at the training phase. During testing, we scale the images to the shorted side of 512. We adopt the simple approach Semantic FPN (Kirillov et al., 2019) as the semantic segmentation method following (Wang et al., 2021b) for fair comparison.

| Method | Backbone | val MS mIoU | Params | FLOPs |
|---|---|---|---|---|
| DANet (Fu et al., 2019a) | ResNet-101 | 45.2 | 69M | 1119G |
| DeepLabv3+ (Chen et al., 2018) | ResNet-101 | 44.1 | 63M | 1021G |
| ACNet (Fu et al., 2019b) | ResNet-101 | 45.9 | - | - |
| DNL (Yin et al., 2020) | ResNet-101 | 46.0 | 69M | 1249G |
| OCRNet (Yuan et al., 2020) | ResNet-101 | 45.3 | 56M | 923G |
| UperNet (Xiao et al., 2018) | ResNet-101 | 44.9 | 86M | 1029G |
| OCRNet (Yuan et al., 2020) | HRNet-w48 | 45.7 | 71M | 664G |
| DeepLabv3+ (Chen et al., 2018) | ResNeSt-101 | 46.9 | 66M | 1051G |
| DeepLabv3+ (Chen et al., 2018) | ResNeSt-200 | 48.4 | 88M | 1381G |
| UperNet (Xiao et al., 2018) | Swin-T (Liu et al., 2021b) | 45.8 | 60M | 945G |
| | AS-MLP-T (Lian et al., 2021) | 46.5 | 60M | 937G |
| | CycleMLP-T (ours) | **47.1** | 60M | 937G |
| UperNet (Xiao et al., 2018) | Swin-S (Liu et al., 2021b) | 49.5 | 81M | 1038G |
| | AS-MLP-S (Lian et al., 2021) | 49.2 | 81M | 1024G |
| | CycleMLP-S (ours) | **49.6** | 81M | 1024G |
| UperNet (Xiao et al., 2018) | Swin-B (Liu et al., 2021b) | 49.7 | 121M | 1188G |
| | AS-MLP-B(Lian et al., 2021) | 49.5 | 121M | 1166G |
| | CycleMLP-B (ours) | **49.7** | 121M | 1166G |

Table 12: The semantic segmentation results of different backbones on the ADE20K validation set.

## F    SAMPLING STRATEGIES

We explore more sampling strategies in this subsection, including random sampling and dilated sampling inspired by dilated convolution (Yu & Koltun, 2016; Chen et al., 2018) (as shown in Figure 6). We also compare the dense sampling method with ours.

**Random sampling.** As shown in Table 13, we conduct experiments with random sampling for three independent trials and observe that the averaged Top-1 accuracy on ImageNet-1K drops by 1.3%. We hypothesize that the decreased performance is caused by the fact that random sampling will totally disturb the semantic information of objects, which is essential to image recognition. Compared with the random sampling strategy, our cyclical sampling is able to aggregate the adjacent pixels, which benefits in capturing the semantic information.

**Dilated Stepsize** (Figure 6). As shown in Table 13, we observe the result of dilated sampling is better than the random one ($+1.0\%$ acc) but lower than ours ($-0.5\%$ acc). In fact, compared with the random sampling, dilated solutions take their advantages in local information aggregation. However, compared with the cyclical sampling strategy, dilated solutions lose the fine-grained information for recognition. It may hurt the accuracy performance to some extent.

**Dense sampling.** we conduct ablation studies by using dense sampling strategies (i.e., vanilla convolution with kernel size $1\times3$ and $3\times1$). Since dense sampling strategies incredibly increase the models' parameters and FLOPs, we do not have enough time to thoroughly optimize the model for 300 epochs. Therefore, for fair comparisons, we conducted extra ablation studies on training models for 100 epochs with the strictly same learning configurations. The results shown in Table 14 demonstrate that the sparse sampling strategy (ours) outperforms the dense one. The comparison indicates that the dense sampling strategies introduce redundant parameters, which makes the model hard to optimize. Our sparse sampling strategy with fewer parameters is proven to be efficient and optimization-friendly.

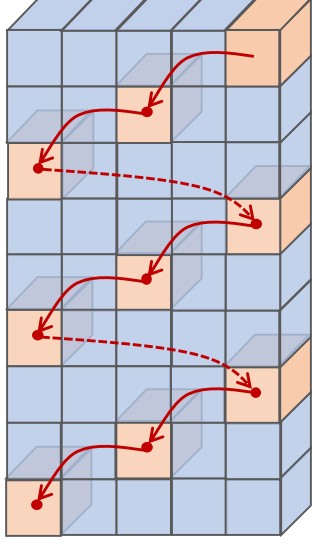

Figure 6: An example of dilated CycleMLP where `dilation=2` and `stepsize=3`.

| Sampling | Params | FLOPs | Top-1 Acc |
|---|---|---|---|
| `dilation=2` | 26.8M | 3.9G | 81.1 |
| Random, `S=1` | | | 80.4 |
| Random, `S=2` | 26.8M | 3.9G | 80.2 |
| Random, `S=3` | | | 80.4 |
| CycleMLP | 26.8M | 3.9G | **81.6** |

Table 13: **Comparison with dilated and random sampling.** For random sampling, we conduct the experiments for three independent trials with three seeds (`S=1, 2, 3`).

| Operators | Dense | Params | FLOPs | Top-1 Acc |
|---|---|---|---|---|
| Conv: $1\times3 + 3\times1$ | ✓ | 34.3M | 5.1G | 75.0 |
| CycleMLP: $1\times3 + 3\times1$ | ✗ | 26.8M | 3.9G | 76.1 |

Table 14: **Comparison with dense sampling:** On the consideration of training time, we only train both models for 100 epochs for fair comparison.

| branch1 | branch2 | ImgNet Top-1 | ADE20K mIoU |
|---|---|---|---|
| $7\times1$ | $1\times7$ | 81.6 | **43.9** |
| $7\times2$ | $2\times7$ | 81.5 | 43.4 |
| $7\times3$ | $3\times7$ | 81.4 | 42.7 |
| $4\times4$ | $4\times4$ | 81.5 | 43.1 |

Table 15: **Comparison on different stepsizes (e.g., even stepsize and odd stepsize)**, including $7\times2$, $4\times4$.

## G   VISUALIZATION EXAMPLES

For easier understanding of our proposed CycleMLP, we visualize several instances of CycleMLP in Figure 7, including general case with stepsize $3\times3$ (7a), even stepsize (7b), and examples where stepsize along `height` or `width` equals to 1 (7c, 7d).

We note that given specific number of input and output channels, no matter how the stepsize changes, the number of parameters of the CycleMLP does not change. Therefore, there is a trade-off of representation abilities between spatial and channel dimensions, which will be discussed in details in following experimental analysis.

**Experiments:** We further conduct experiments on CycleMLPs with stepsize of $2\times7$, $7\times2$, $7\times3$, $3\times7$, and $4\times4$, respectively. The results are summarized Table 15. For fair comparisons, all the models in the above table have the same parameters and FLOPs. We observe that the model with stepsize of $1\times7$ and $7\times1$ achieves the best performance, especially for semantic segmentation on ADE20K. To analyze the impact of stepsize on the performance, we take Figure 7 for better illustration. One can see that enlarging the stepsize can expand the spatial receptive field. However, at a cost, it will reduce the number of periods (groups) running along the channel dimension, which may hurt the channel-wise representation abilities. Taking a feature map with $C = 18$ for example, the CycleMLP with stepsize $3\times3$ (Figure 7(a)) runs through only 2 channel groups (curly brackets in the figure). However, the CycleMLP with stepsize $3\times1$ (Figure 7(c)) will run through 6 groups in total, making better use of the representation in the channel dimension. That's to say, there is a trade-off between spatial and channel representation. We empirically found that CyCleMLP with stepsize of $1\times7$ and $7\times1$ achieves the best performance.

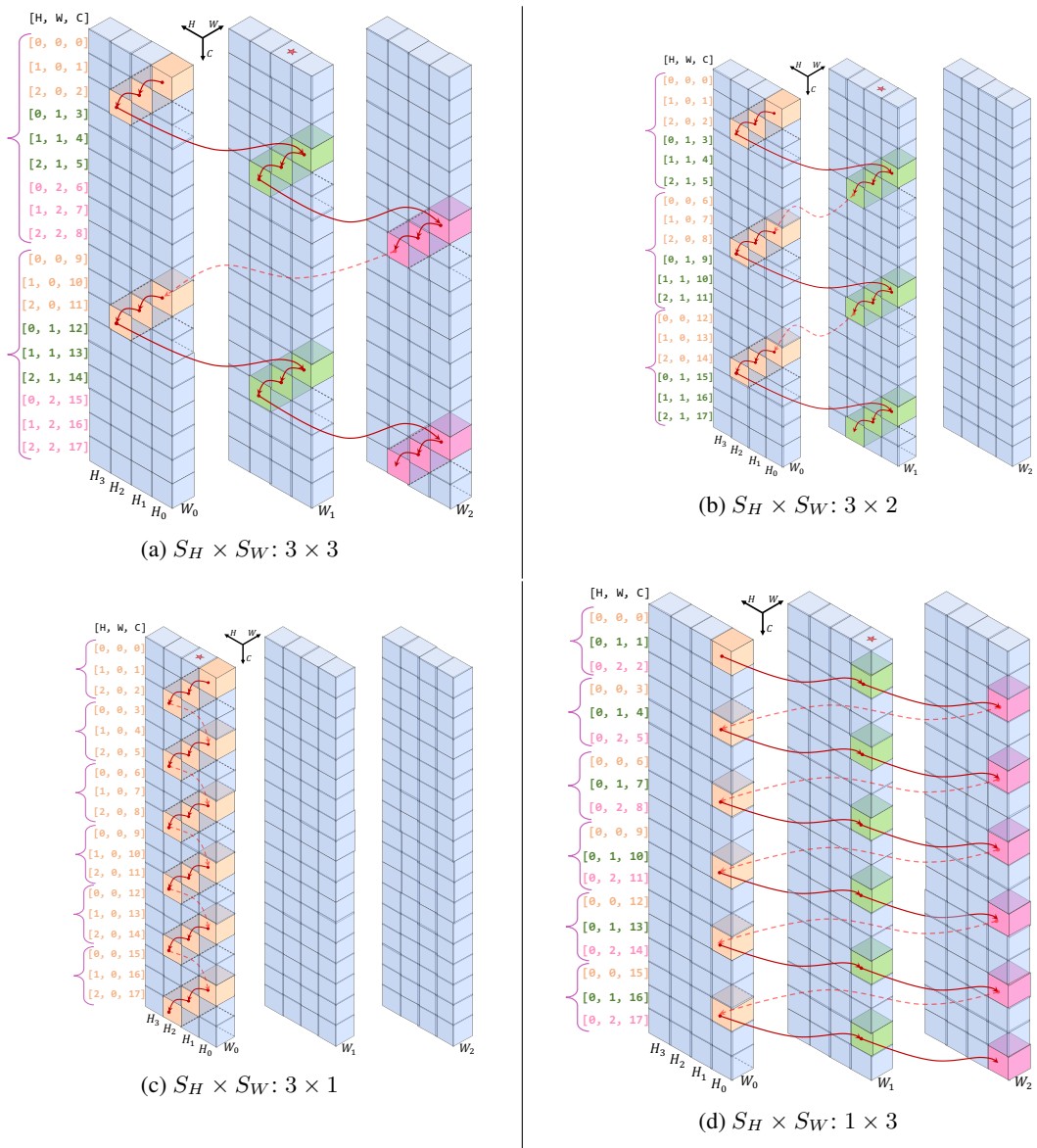

Figure 7: **Examples of `Stepsize` cases:** Here we separate the feature map along the `width` dimension for convenient visualization. ★ denotes the output position. We place the absolute coordinates (`h, w, c`) of the sampled points at the left of the feature. Sampled points within a curly bracket ({) belong to the same period (group). Dash lines link two cyclical periods.

