# OpenReview forum: "CycleMLP: A MLP-like Architecture for Dense Prediction"
_ICLR.cc/2022/Conference — ICLR 2022 Oral_

### Official Review · Reviewer_34rq · 2021-11-02

**Correctness:** 4
**Technical Novelty And Significance:** 3
**Empirical Novelty And Significance:** 3
**Recommendation:** 8
**Confidence:** 3

**Main Review:**

Strengths: 1. The Cycle FC is capable of dealing with various image scales, and has linear computational complexity the same as channel FC and a larger receptive field than Channel FC. 2. CycleMLP achieves competitive results on object detection, instance segmentation, and semantic segmentation.

Weaknesses: 1. The Cycle FC align features at different spatial locations to the same channel, but analysis is slightly insufficient. There could be many different designs of it. For example, the experiments or analysis with different sampling intervals and sample size. 2. CycleMLP is slightly insufficient in discussion of the design and ablation studies.

**Summary Of The Paper:**

This paper presents a  MLP-like architecture, CycleMLP. The proposed CycleMLP can be used for dense prediction, which is more suitable for object detection or image segmentation tasks.  The experiments show the effectiveness of the proposed model. The proposed Cycle FC can reduce the amount of network parameters and calculation, and is insensitive to image resolution.

**Summary Of The Review:**

The proposed method is simple and interesting. Compared to modern MLP architectures, the CycleMLP can cope with various image sizes and achieves linear computational complexity. The benefits of each component and the performance difference are shown. All experiments of this paper is detailed and the overall content is sufficient. The proposed method is applicable to many vision tasks in the future.

---

### Official Review · Reviewer_4cNH · 2021-11-02

**Correctness:** 3
**Technical Novelty And Significance:** 2
**Empirical Novelty And Significance:** 3
**Recommendation:** 6
**Confidence:** 4

**Main Review:**

Pros:
(1)  This paper designs a new MLP-like architecture that can cope with various image sizes and achieves linear computational complexity.
(2)  The authors contribute amounts of experiments to show the comparable performance even higher performance in semantic segmentation.

Cons:
(1)  It is a little difficult to understand Figure 1. In Figure 1(c), you implement cycle operation in the HW dimension, but in Figure 1(d), it seems that you only implement cycle operation in the H dimension. It needs to be clarified.
(2)  In Table 4 and Table 5, the authors conduct ablation studies about stepsize. It is implemented when stepsize is 1x7 and 7x1, what if the stepsize is 2x7 or 7x2? Does this situation exist? Another, can the stepsize be an even number?
(3)  Besides, ViP uses the 3-branch structure and AS-MLP seems to be the concurrent work. What are the specific differences between CycleMLP and these architectures?
(4)  About Figure 4, what means the effective receptive field? And what does the higher color value mean? From this figure, Swin has the sharp ERF and CycleMLP has smoother ERF. Can you explain it?


**Summary Of The Paper:**

This paper presents a simple MLP-like architecture, CycleMLP for image classification, object detection and segmentation. It can cope with various image sizes and achieves linear computational complexity to image size. CycleMLP achieves competitive results on object detection, instance segmentation and semantic segmentation. Further, it outperforms Swin-Tiny on the ADE20K dataset with fewer FLOPs.

**Summary Of The Review:**

This paper shows competitive results of MLP-like architecture on image classification, object detection and segmentation. However, there exist some unclear descriptions. Currently I rate it as a borderline paper and the authors need to consider the above questions.

---

### Official Review · Reviewer_dMw3 · 2021-11-03

**Correctness:** 3
**Technical Novelty And Significance:** 3
**Empirical Novelty And Significance:** 3
**Recommendation:** 8
**Confidence:** 3

**Main Review:**

The paper is strong in the sense that it proposes a simple yet effective idea to extend previous MLP work for image tasks.  In a sense the basic idea of a sparse sampling across spatial and channel space is a natural variation to the other extremes in previous work such as channel and spatial FC.  Thus the methodology follows the basic pattern as previous work so doesn't require much explanation in terms of details.  In any case, the primary contribution here is the extensive sense of experimental results.  Figure 2 and Table 2 already provide decent support for CycleMLP, and the results showing FLOPs and parameter counts further strengthens the results.  Also as CycleMLP should be generic in terms of tasks the additional results on detection and segmentation make the idea more convincing.  Overall the basic contribution of improving MLP models to have linear complexity and applicability to varying image sizes is solid.

Although the main work here is experimental in proving out the idea, it would be useful to have a deeper analysis of why the method works well.  In particular, the strided sampling of CycleMLP seems a bit motivated by ease and computational complexity, but seems it doesn't quite have the basic property of CNNs in being adapted to local correlations in natural images.  Presumably the patch embedding part is critical here as it already explicitly enforces spatial locality before CycleMLP is applied?  Do the previous MLP methods use patch embedding, and if they do, would they also have similar performance?  In other words, how much of the desired properties of linear computational complexity and applicability to varying image sizes comes from the patch embedding itself?

**Summary Of The Paper:**

This paper proposes an improvement to existing MLP style models for image tasks that extends the general MLP competitive work compared to CNN models while attempting to capture some desirable CNN properties such as applicability to varying image sizes.  In addition the method has linear computational complexity as compared to previous MLP work which have quadratic complexity.  The basic idea is a variant of previous channel and spatial FC ideas by having a node's receptive field sample across spatial and channel domains rather than being fixed to either just channel or spatial aggregation.  The overall architecture uses the previous work on patch embeddings and idea of stages with stacked blocks.  Comprehensive experimental results are show improvement over previous MLP models and competitive results to other methods such as CNNs and transformers on multiple tasks including classification and segmentation.

**Summary Of The Review:**

Overall the comprehensive and varied experimental results give the paper its strength.  The results are either competitive, such as on segmentation, or better than the previous MLP methods.  The idea is simple and a natural extension of previous ideas for fully connected layers.  There is a question on the relative contribution of the patch embedding when comparing to previous MLP methods, but generally no major weaknesses in the paper.

---

### Official Review · Reviewer_QA1s · 2021-11-03

**Correctness:** 3
**Technical Novelty And Significance:** 3
**Empirical Novelty And Significance:** 3
**Recommendation:** 8
**Confidence:** 4

**Main Review:**

The method modifies the existing 1x1 convolution (channel FC in paper), to sample points in a cyclical manner inside neighbourhood window along the channel dimension. It has the same weight shape as 1x1 convolution, ie: C_in x C_out. Stepsize is used to control the receptive field and can be thought of as analogous to kernel size in convolution. Overall Cycle MLP mimics the effect of sparse convolution kernel where each element along channel dimension is sampled at a different spatial position in a cyclical manner.

Strengths:
- Simple idea with promising results.
- Readily adaptable to existing architectures with vanilla MLP.
- Increases the receptive field without increasing the number of  parameters

Weakness/suggested improvements:
- There are no ablations or explanations on how/why the induced sparsity is helping in performance.
- It is not evident that why cyclical sampling is better than random sampling or what is the significance of cyclical sampling. Some discussion and experiments around this point would be useful.
- A minor typo: MHSA is written as MSHA below eqn3



**Summary Of The Paper:**

The paper proposes Cycle MLP architecture, the idea is to bring spatial context into Channel FC and increase its receptive field.  The main objective of the paper is to address the challenges faced by the current MLP-Architectures. Cycle MLP allows flexible image resolution and avoids quadratic computational complexity in dense architectures. The paper presents results in a variety of tasks and shows demonstrates promising performance.


**Summary Of The Review:**

Overall a simple idea and seems to work well. However, some more intuition around cyclic sampling would be worthwhile and might improve the paper. Exploring alternate sampling strategies would be interesting (random sampling or some other pattern can be explored).

---

### Decision · Program_Chairs · 2022-01-20

**Decision:**

Accept (Oral)

**Comment:**

The authors propose a new MLP-Mixer-like architecture called Cycle MLP which has two main advantages with respect to MLP-Mixer: (i) it’s applicable to varying input image sizes, and (ii) linear computational complexity. The authors present competitive results on image classification, object detection and segmentation.

The reviewers felt that both (i) and (ii) are key issues in the current MLP-Mixer-based models. The reviewers also appreciated the simplicity of the idea and the execution of the empirical evaluation. During the rebuttal and discussion phase the authors provided compelling evidence for the issues pointed out in the initial review.

Given that MLP-Mixer-based architectures are becoming increasingly popular, I believe that these contributions will be of great interest to the ICLR community and I will recommend acceptance.